# Advances in the Immunomodulatory Properties of Glycoantigens in Cancer

**DOI:** 10.3390/cancers14081854

**Published:** 2022-04-07

**Authors:** Valeria da Costa, Teresa Freire

**Affiliations:** Laboratorio de Inmunomodulación y Desarrollo de Vacunas, Departamento de Inmunobiología, Facultad de Medicina, Universidad de La República, Montevideo 11800, Uruguay; valedacosta21@gmail.com

**Keywords:** glycans, cancer, tumour-associated carbohydrate antigens, C-type lectin receptors, immunomodulation, metastasis

## Abstract

**Simple Summary:**

This work reviews the role of aberrant glycosylation in cancer cells during tumour growth and spreading, as well as in immune evasion. The interaction of tumour-associated glycans with the immune system through C-type lectin receptors can favour immune escape but can also provide opportunities to develop novel tumour immunotherapy strategies. This work highlights the main findings in this area and spotlights the challenges that remain to be investigated.

**Abstract:**

Aberrant glycosylation in tumour progression is currently a topic of main interest. Tumour-associated carbohydrate antigens (TACAs) are expressed in a wide variety of epithelial cancers, being both a diagnostic tool and a potential treatment target, as they have impact on patient outcome and disease progression. Glycans affect both tumour-cell biology properties as well as the antitumor immune response. It has been ascertained that TACAs affect cell migration, invasion and metastatic properties both when expressed by cancer cells or by their extracellular vesicles. On the other hand, tumour-associated glycans recognized by C-type lectin receptors in immune cells possess immunomodulatory properties which enable tumour growth and immune response evasion. Yet, much remains unknown, concerning mechanisms involved in deregulation of glycan synthesis and how this affects cell biology on a major level. This review summarises the main findings to date concerning how aberrant glycans influence tumour growth and immunity, their application in cancer treatment and spotlights of unanswered challenges remaining to be solved.

## 1. Tumour-Associated Carbohydrate Antigens

Glycosylation is a highly regulated and complex posttranslational modification that is of great importance in the modulation of the activities and functions of glycoproteins in biological systems. The mammalian glycome repertoire is estimated to be between hundreds and thousands of glycan motifs, with a vast diversity of glycan structures [1]. These are a result of different linkages and carbohydrate moieties that combine to generate different structures, which encode various types of information and allow them to participate in a great number of cellular processes, such as cell proliferation, adhesion, and migration. These effects are mediated by the glycan charge, bulkiness, and hydrophilicity, as well as the capacity of transmitting information by interacting with specific lectin or lectin-like receptors [2].

Fundamental changes in the glycosylation patterns of cell surface and secreted glycoproteins occur during malignant transformation and cancer progression [1,3]. Changes in glycosylation induce the appearance of unmasked glycan antigens in the surface of tumour cells, known as tumour-associated carbohydrate antigens (TACAs) [3,4,5,6]. There is a vast amount of documented evidence that changes in glycosylation are a hallmark of cancer and their role in tumour development has been extensively studied [3,6].

Aberrant protein glycosylation in cancer includes both changes in *N*-glycosylation and *O*-glycosylation. The expression of cancer-associated glycans, such as sialyl-Lewis^x^ (sLe^x^), Thomsen-nouvelle antigen (Tn), and sialyl-Tn (sTn) antigens has been detected in almost every human carcinoma cancer type [7]. The glycan structures of these glycoantigens are depicted in Figure 1.

In this review we will first focus on the different glycans structures found in tumours, the biological effect of aberrant glycosylation in tumour cells and how glycans may modulate cancer immunity by interacting with specific receptors found in immune cells.

### 1.1. N-glycan Branching

*N*-glycosylation is characterised by the addition of a precursor glycan strcture onto the side chains of Asn (within the consensus sequence Asn-X-Ser/Thr), which is then decorated by the action of both glycosidases and glycosyltransfersases [8]. TACAs in *N*-glycans can also comprise increased formation of extensively branched structures [9]. This is mainly catalysed by three glycosyltransferases, which will be described below.

The β1,4-*N*-acetylglucosaminyltransferase (GnT-III) catalyses the transfer of GlcNAc from UDP-GlcNAc to biantennary sugar chains to produce a β1,4GlcNAc linkage designated as a bisecting GlcNAc structure [10]. Interestingly, *N*-glycans with a bisecting structure were shown to decrease with the development of malignancy in colorectal cancer [11].

On the other hand, the increased expression of β1-6 branched *N*-oligosaccharides, catalysed by the golgi enzyme *N*-acetylglucosaminyl transferase-V (GnT-V), is also related to cancer progression and aggressiveness, particularly in the breast, colon, oesophagus gliomas and endometrium [12,13]. The β1,6-*N*-acetylglucosaminyl transferase (GnT-V) catalyses β1,6GlcNAc branching, which may later be elongated to a polylactosamine and a sLe^x^ structure on various growth factor receptors such as epidermal growth factor receptor (EGFR), fibroblast growth factor receptors (FGFR) and Insulin Like Growth Factor 1 Receptor (IGFR) on the cell surface [10].

Finally, the α1,6-fucosyltransferase (FUT8) catalyses the transfer reaction from GDP-Fucose to produce a core fucose moiety [9]. Alterations in its expression have been reported in hepatoma, pancreatic, colon, liver, breast, prostate, thyroid non-small cell lung cancer and melanoma [10].

In a recent proteomic study of *N*-glycosylation in breast cancer tissues, Scott and collaborators detected an increased number of Man8/Man9 glycans in HER2^+^ tissues [14]. This is an indicator of less mannosidase processing during biosynthesis, which leads to proteins that are normally destined for lysosomes to be secreted or transported to the cell surface [14]. In the same study, the authors evidenced that the presence of poly-LacNAc structures (a linear structure comprising the repeating *N*-acetyllactosamine unit (Galβ1-4GlcNAcβ1-3) (see Figure 1) are associated with more aggressive breast cancers [14].

### 1.2. Mucin Type O-glycans

*O*-linked glycosylation is characterised by the addition of glycans onto the side chains of Ser or Thr [8] on a one step-manner process that includes the addition of single monosaccharides in sequential order by glycosyltransferases [15]. Several tumour-associated *O*-glycans have been identified and extensively characterised as key modulating agents of tumour progression. These tumour *O*-glycans comprise oncofetal antigens (which are rare in normal adult tissue but may be found in embryonic tissues) neoantigens (which are not found neither in adult or in embryonic tissues and therefore conform novel structures), and antigens expressed in altered levels [16]. The most common TACAs formed from incomplete *O*-glycan synthesis are GalNAcα-*O*-Ser/Thr (Tn, Thomsen Nouveau, CD175), Neu5Acα2,6-GalNAcα-*O*-Ser/Thr (sTn, sialyl-Tn, CD175s), Galβ1,3-GalNAcα-*O*-Ser/Thr (TF, Thomsen-Friedenreich, CD176, also known as T antigen) and Neu5Acα2,6- and Neu5Acα2,3-Galβ1,3-GalNAcα-*O*-Ser/Thr (2,6-sTF, 2,3-sTF, respectively) [17]. In addition, some extended *O*-glycans may be found in tumour cells, such as the ABO(H) antigens, the Lewis antigens and poly-*N*-acetyllactosamine [16] (Figure 1).

Often, mucins, which are large and heavily O-glycosylated proteins, have been associated with aberrant *O*-glycosylated antigens [17]. In addition, their expression in cancer cells is reported to be associated with a more malignant phenotype. For instance, Mucin-1 (MUC1) expression on tumour cells differ from normal cells, mainly due to an upregulation of gene expression and to *O*-glycans being truncated, mostly exhibiting core 1 *O*-glycan structures [17]. Moreover, MUC1 is a prognostic marker for colorectal adenocarcinoma [18], while MUC4, MUC1 and MUC5A are reported as biomarkers for pancreatic cancer [19,20,21]. In addition, MUC16, which is overexpressed in ovarian cancer and pancreatic cancer cells, enhances cell survival, mainly by protecting them from recognition by cytotoxic cells such as Natural Killer (NK) cells, and enabling their dissemination and development of metastasis in the peritoneal cavity [22]. MUC16 from metastatic pancreatic cells can also bind to E- and L-selectin mediated by sialofucosylated structures in the mucin *O*- and *N*-glycans, enhancing their metastatic capacity [23]. Furthermore, MUC4 in breast cancer is associated with circulating tumour cells and with an increase of tumour cell survival in circulation, promoting metastasis [24]. Nevertheless, the expression of mucins may also be associated with a protective phenotype towards malignancy. For instance, MUC2 acts as a protective component of the mucus layer in the colon. Indeed, MUC2 downregulation is associated with tumour progression in colon cancer and metastasis, via the enhancement of the proinflammatory axis IL-6-gp130-STAT3, that contributes to chronic inflammation and tumorigenesis [25].

### 1.3. Sialylation

Sialylation is a relevant modification in cellular glycosylation, as sialic acids are known to participate in cell signalling, immune response and cellular interaction [26]. Sialic acids have a negative charge at physiological pH, which affects protein conformation and oligomerization, as well as interaction with other proteins or components of the extracellular matrix [27]. There are twenty Golgi localised sialyltransferases with different substrate and linkage specificity in charge of protein and lipid sialylation [27]. The most common types of sialic acids in mammals are *N*-acetylneuraminic acid (Neu5Ac) and *N*-glycolylneuraminic acid (Neu5Gc), but humans lost the ability to synthesise Neu5Gc due to a gene deletion in the enzyme in charge of sialic acid synthesis [27]. Nevertheless, there still can be found Neu5Gc in human tissues, obtained from diet, such as red meat, and the incorporation of this to glycoproteins and gangliosides has been shown to promote inflammation and tumour progression [28].

Cancer-associated glycans often exhibit an increased amount of sialic acid. Augmented sialylation of tumour cells has been correlated with a metastatic phenotype and poor prognosis in patients with cancer [29]. Both *N*- and *O*-glycans have terminal sialic acid incorporated in their structures. Some relevant sialylated TACAs include the α2,6-sialylated lactosamine (Sia6LacNAc), synthesised by the β-galactoside α2,6-sialyltransferase I (ST6Gal-I), an enzyme with altered expression in various types of cancer, such as colon, stomach, and ovarian cancer [30]. Furthermore, the sialylated variants of sLe^a^ and sLe^x^ are highly expressed in cancer and correlate with decreased survival [30]. In the same line, the sulphated sLe^a^ and sLe^x^ variants and nonfucosylated sialyl-LacNAc type [31] have been associated with particular differentiation states and subgroups of pancreatic cancer cells [32].

### 1.4. Glycosphingolipids

Lipid glycosylation in the secretory pathway gives place to glycosphingolipids (GSLs), including the glycolipid that anchors glycosylphosphatidylinositol (GPI)-linked proteins [1]. Glycolipids are compounds containing a carbohydrate group linked by glycosyl linkage to a lipid moiety [33]. The basic structure of the GSL is the ceramide, determined by long-chain amino alcohol sphingosine, originated from palmitoyl coenyme-A and Ser, in amide linkage to a fatty acid. Monosaccharides or oligosaccharides chains are ligated to the ceramide via glycosidic linkage, constituting the basic platform for chain extension to the oligoglycosylceramides [33].

The main changes in cancer progression derive from differences in GSL level expression. Gangliosides such as GM3, GM2, CD3 and GD2 are often found overexpressed in tumours compared to normal tissues, including lung cancer, melanoma, and neuroblastoma [26].

### 1.5. Glycosaminoglycans

Glycosaminoglycans (GAGs) are also linked to serine and threonine; they are linear, constructed through disaccharide repeats formed by GlcNAc or GalNAc, combined with an uronic acid or galactose, and are produced by different biosynthetic pathways than *N*- and *O*-glycans and are often highly sulfated [1,34]. Glycosaminoglycans are functionally diverse and include heparan sulfate, chondroitin sulfate and dermatan sulfate [35]. The biosynthesis of GAGs begins in the cellular cytoplasm and continues in the Golgi apparatus, where sulfation of functional groups takes place. Then, the sulfated GAGs are linked covalently to anchor proteins called proteoglycans [36]. Of note, hyaluronic acid does not go through sulfation in Golgi apparatus, but the precursor sugar is transported to the cellular membrane for further processing without sulfation [36].

Interestingly, low levels of cell surface heparan sulphate and syndecan-1 correlate with high metastatic activity of many tumours, including colon, mesothelioma, lung, hepatocellular carcinoma, infiltrating ductal carcinoma of the breast, and head and neck carcinoma [37]. In colorectal cancer, it has been reported that neoplastic tissues have an increased amount of chondroitin sulphate and dermatan sulphate compared to non-neoplastic mucosal tissues, while heparan sulphate was decreased in neoplastic tissues [38]. Furthermore, it has also been observed that the alteration in the expression of heparan sulphate proteoglycans is dependent on the metastatic nature of the tumour [39].

## 2. TACAs as Mediators of Cancer Progression

The cellular surface exposes a great variety of glycoconjugates, comprising macromolecules containing carbohydrates in their structure. The membrane asymmetry exposes the glycan portion of the proteins to the extracellular space, forming, therefore, the glycocalyx [40,41]. This lightens, as already expressed, the importance of glycans in the normal cellular processes, such as cell–cell communication, interaction with the extracellular matrix and soluble factors. Both the glycans in the surface proteins and the ones expressed in soluble proteins play a fundamental role in modulating and mediating several key events in the development and progression of cancer [42]. Furthermore, some works demonstrate their role as potential biodiagnostic markers of cancer, to follow up patient prognosis and to develop antitumor immunotherapy strategies [43,44]. To illustrate some examples, detecting a glycan signature in serum prostate specific antigen (PSA) increases the sensitivity of this approach as a diagnostic tool for pancreatic cancer [45]. Furthermore, the detection of ⍺-2,3-sialic acid in PSA enhances sensitivity to differentiate high-risk malignant cancer from benign disease [46]. Serum sialyl-sTn levels are also reported to be associated with histological grade and lymph node metastasis in endometrial cancer [47]. Moreover, a study showed that sialyltransferase levels in plasma of cancer patients are elevated [48].

### 2.1. Cellular Proliferation and Survival

Glycans may stabilise the expression of cellular receptors on the cell surface and therefore modulate the interaction between the receptor and its ligand, and its downstream signalling. For instance, the role of *N*-glycans is crucial in checkpoint signalling mediated by the programmed death-1 (PD1) and PD1 ligand (PD-L1). PD-L1 is reported to be glycosylated in various types of cancer, including melanoma, breast, lung, and colon [49]. In triple negative breast cancer, the downregulation of the β-1,3-*N*-acetylglucosaminyl transferase (B3GNT3) enhances T cell mediated anti-tumour immunity mediated by PD-L1 and PD1 interaction [50]. Furthermore, *N*-glycans may also modulate growth factors receptors signalling, including EGFR, platelet-derived growth factor (PDGF), hepatocyte growth factor receptor (HGFR), FGFR, and IGFR [51], mainly due to the high number of *N*-glycan sites in growth factors. The receptors bind to each other, forming a grid making them resistant to endocytosis, which enhances active signalling that leads to cell growth and proliferation in cancer [10]. In addition, high mannosylated glycoforms are expressed in EGFR, comprising a strong tumour marker [52].

Furthermore, ceramide glycosylation in glycosphingolipids participate in the cell signalling through the proto-oncogene tyrosine-protein kinase Src/β-catenin pathway and participate in the maintenance of pluripotency in undifferentiated embryonic stem cells [53]. This property has been exploited by breast cancer stem cells, in which glycosylation and globotriaosylceramide (Gb3) expression enhancement correlate with the number of these cells. Silencing the enzyme responsible for the synthesis of this glycan moiety, the glucosylceramide synthase, results in the death of the breast cancer stem cells through the deactivation of c-Src/β-catenin signalling [53].

Proteoglycans are also involved in cell signalling. Particularly, Syndecan 1, a cell surface heparan sulphate proteoglycan, acts as a coreceptor for a multitude of biological factors such as growth factors, angiogenic factors, cytokines, and chemokines [54]. In cancer, the dysregulated expression of Syndecan 1 alters cell proliferation and invasion [54].

In some cases, aberrant glycosylation may potentiate intrinsic protein malignant features. *O*-glycosylated MUC4, a glycoprotein overexpressed in pancreatic ductal adenocarcinoma (PDAC), potentiates malignancy properties in cancer cells [55]. Interestingly, it has been seen that glycosylation enhances MUC4 malignant properties through the interaction with epidermal growth factor receptors, and reduces sensitivity to gemcitabine, via altering ErbB/AKT signalling cascades and expression of nucleoside transporters [55]. Finally, hypersialylation of tumour cell surfaces also enhances cancer resistance to chemotherapy [27].

### 2.2. Adhesion, Migration and Metastasis

Glycans participate in cellular adhesion and allow the reorganisation of the cell surface to expose proteins essential for cellular communication and interaction. For instance, mucins interact with cell-adhesion molecules, and can block cell–cell adhesion mediated by E-cadherin and integrins. E-cadherin, a transmembrane glycoprotein, is the main epithelial cell–cell adhesion molecule, and loss of its expression or mislocalization to cytoplasm is related to the epithelial-to-mesenchymal transition (EMT), wound healing, cancer progression, and metastasis [56,57]. Particularly, O-GalNAc glycosylation of E-cadherin mediated by GalNac-transferase 3 (GALNT3) correlates with the localization of E-cadherin, and when GALNT3 expression is lost, E-cadherin is retained in the Golgi apparatus, which disrupts adhesion junctions and induces EMT [58]. Selectins are glycan binding proteins that bind TACAs present in the surface of glycoproteins or in the extracellular matrix of cancer cells. They are vascular adhesion molecules expressed on endothelium that can promote cancer metastasis and tumour growth in distant organs [59,60,61,62]. They also participate in leukocyte trafficking and their recruitment to inflammatory sites [63]. Modifications in *O*-glycans, for instance silaylated Lewis structures in colon cancer, that are ligands for E-selectins enhance tumour invasion and metastasis [64]. Furthermore, sTn has been shown to have implications in cell growth and migration. While overexpression of sTn seems to slow cell growth, it increases cell migration and therefore enhances metastasis [65,66]. In addition, the expression of sLe^x^ antigen in gastric cancer cells induces an increased invasive phenotype through the activation of tyrosine-protein kinase Met, in association with proto-oncogene tyrosine-protein kinase Src, focal adhesion kinase (FAK), cell division control protein 42 homolog (Cdc42), ras-related C3 botulinum toxin substrate 1 (Rac1) and ras homologous protein (RhoA) GTPases activation [67].

Interestingly, ⍺1-6fucosylation of *N*-glycans in TGFβ catalysed by FUT8 stimulates breast cancer invasion and metastasis [68] promoting the EMT process [69,70]. In addition, GnT-III activity can also drive stem cell expansion to promote growth and invasion of cancer cells by EMT that depends on Notch, Wnt, and TGFβ signalling [71]. On the other hand, a microRNA-198-5p that targets FUT8 in non-small cell lung cancer cells inhibits cell migration, invasion, EMT and in vivo metastases [72]. EMT induction also promotes changes in sialylation that alter physico-chemical properties of proteins and allow malignant cells to liberate from tumours and migrate to invade other tissues. For example, when EMT occurs in colon cancer, sialyltransferases expression is upregulated, particularly those involved in the synthesis of sLe^x^ and sLe^a^, that serve as ligands for E-selectin [27]. Moreover, the experimental induction of EMT with TGFꞵ in prostate epithelial cells leads to increased levels of oncofetal fibronectin, with an increased expression of Tn and core 1 structures. In fact, when the glycosyltransferases that mediate the synthesis of these antigens were knocked out, the induction of oncofetal fibronectin by TGFꞵ was inhibited and the migratory cell potentiation was reversed [73].

Heparan sulphate proteoglycans (HSPGs) have also been shown to promote cell-to-cell and cell-to-extracellular matrix adhesions, therefore inhibiting invasion and metastasis [74]. On the other hand, the decrease of HSPG levels, as seen in some cancers, results in the malignant cells being more invasive [37].

## 3. Mechanisms Altering Glycan Synthesis

Glycan synthesis occurs in a highly regulated sequential or competitive action of different glycosyltransferases and glycosidases [1,3]. Glycosyltransferases synthesise glycan chains, whereas glycosidases hydrolyze specific glycan linkages [1,6]. The expression, activity and subcellular location of these enzymes dictate the overall glycosylation profile in cells or tissues [3]. Since protein glycosylation has a high impact on a wide range of cell biological processes, as described above, changes in glycosidase or glycosyltransferase expression, subcellular localization or activity dictate the glycan motives present in a cell, and can thus modify cell proliferation, adhesion, differentiation and immune responses [1,30,41,42]. While the incomplete synthesis is a phenomenon that occurs in the early stages of cancer progression and leads to the appearance of truncated forms of *O*-glycans (such as Tn or sTn antigens), the neosynthesis takes place in advanced stages of neoplasia and is associated with the induction of genes involved in carbohydrate synthesis, and derives in the formation of de novo antigens such as sLe^a^ or sLe-^x^ [1,30,41,42,75].

A large amount of evidence has demonstrated that incomplete *O*-glycan TACA expression is a consequence of a dysregulation of the initial steps of *O*-glycan biosynthesis mediated by polypeptide-*N*-acetylgalactosaminyl-transferases (ppGalNAcTs) that transfer UDP-GalNAc to Ser/Thr residues in a polypeptide, forming Tn antigen (GalNAc1-Ser/Thr) [17,30]. Indeed, an overexpression, a higher activity or a modified subcellular location of these transferases, as well as defects in the T-synthase or its associated chaperone Cosmc, can lead to higher Tn antigen expression levels [75,76,77,78,79,80,81,82]. On the other hand, an upregulation of GalNAc2,6-sialyltransferase (ST6GalNAc-I) in tumours results in the overexpression of sTn, (NeuAc2,6-GalNAc-Ser/Thr) preventing further *O*-glycan elongation. Indeed, increased activity or expression of sialyltransferases leads to the hypersialylation of cell surfaces, which as stated before is one of the most common glycosylation changes that occurs in tumours [48]. Furthermore, an upregulation of the TF antigen (Gal1,3-GalNAc 1-*O*-Ser/Thr) can be the consequence of upregulation of T-synthase [83]. Interestingly, it would seem that hormone therapy of breast cancer cells and oxidative stress induced the expression of incomplete TACAs in *O*-glycans [84] through a process that still remains to be understood. These cancer-associated glycoantigens are frequently decorating mucins on epithelial tumours.

With respect to *N*-glycan associated TACAs, the overexpression of fucosyltransferases (Fuc-T 1/3) increases terminal fucosylation leading to the expression of Le^x^ and Le^y^ [85]. Furthermore, the upregulation of sialyltransferases leads to sialylated species of these antigens (sLe^x^ and sLe^y^), while increased activity levels of the mannoside acetyl-glucosaminyltransferase 5 (MGAT5) increases *N*-glycan branching [86]. Furthermore, dysregulation of the 4-*N*-acetylgalactosaminyltransferase 3 (ꞵ4GALNT3) leads to the expression of terminal LacdiNAc (GalNAc1,4GlcNAc1-) on both *N*-linked and *O*-linked glycans [87,88]. Finally, an upregulation of different potential polypeptide glycosylatransferase substrates can also upregulate the expression of TACAs in cancer [78,79,82]. Therefore, these reports indicate the complexity of the molecular processes underlying TACA expression in cancer since multiple glycosyltransferases and glycosidases, as well protein expression are involved.

## 4. Tumour-Associated Glycans Possess Immunomodulatory Properties

The immune system is capable of recognizing malignant cells and inducing effective antitumor immunity able to eradicate cancer cells [89]. The capacity of dendritic cells (DCs) to sense tumour-associated antigens and stimulate naive T cells enables them to initiate the adaptive immune response [90,91]. CD4^+^ T cells are activated via recognition of exogenous derived antigens loaded in major histocompatibility complex class II (MHC II), while CD8^+^ T cells are activated via endogenous derived-antigen loaded in MHC class I (MHC I) by antigen presenting cells, as well as via exogenous-derived antigens loaded in MHC I in a process known as cross-presentation performed by DCs [92,93]. Differentiation of CD4^+^ T cells into T helper-type 1 (Th1) is crucial for the antitumor immune response since they promote IFNℽ-mediated cytotoxic mechanisms driven by other cells [94]. Indeed, both cytotoxic CD8^+^ T cells (CTLs) and their innate counterpart, NK cells, can recognize tumour cells and mediate specific killing by cytotoxicity [95]. Apart from NK cells and DCs, other innate cells such as inflammatory M1 macrophages play important roles in tumour eradication [95,96,97].

Nevertheless, aggressive tumours develop immune evasion mechanisms that protect them from the antitumour immune response by regulating specific effector immunity or inducing other processes such as angiogenesis [98]. Indeed, regulatory T cells (Treg), myeloid-derived suppressor cells (MDSCs), tumour-associated macrophages (TAM) and regulatory DCs, among others, interfere with antitumor immunity and promote tumour growth and metastases development [99]

DCs and TAMs play an essential role in promoting tumour development, being predominantly polarised in the tumour microenvironment to regulatory DCs or alternative activated or M2 macrophages, with anti-inflammatory and proangiogenic functions [100,101]. M2 TAMs also facilitate tissue remodelling and tissue repairing [102]. In addition, the tumour microenvironment is characterised by the expression of immunoregulatory molecules, which are expressed either in cancer cells or leukocytes. Indeed, PD-L1 or PD-L2 expression by tumour cells can inhibit antitumor immunity by interacting with PD1 expressed on cytotoxic T cells and NK cells [97,99,103], causing T cell anergy and inhibit tumoricidal capacity of CTLs [,[104]]. Cytotoxic T-lymphocyte-associated protein 4 (CTLA-4) is another crucial molecule involved in immune regulation in cancer. CTLA-4 is expressed on Tregs and competes with CD28 for binding to costimulatory molecules (CD80 and CD86) on antigen-presenting cells, thereby inhibiting the activation of T cells [105]. Thus, these molecules play a key role in immune regulation and peripheral tolerance [106,107], and their blocking or signalling inhibition constitute one of the most studied antitumor immunotherapy strategies currently [108,109,110,111,112]. Finally, recent reports have demonstrated that the immunological characteristics in the tumour microenvironment also modulate the metabolism of both tumour cells and innate leukocytes, specially DCs and macrophages [113].

### 4.1. TACAs as Immunomodulators of Antitumor Immunity

Glycan signatures on tumour cells can be recognized by the immune system. In particular, C-type lectin receptors (CLRs) expressed on a variety of leukocytes mediate the interaction between TACAs and the immune system. CLRs consist of a family of one type of transmembrane pattern recognition receptors (PRRs) that recognize glycosylated pathogen-associated molecular patterns (PAMPs) or glycoantigens expressed on tumour cells, among others [114,115,116,117]. While CLRs recognise their ligands in a Ca^2+^-dependent manner, some alternative C-type lectin-like receptors do not need Ca^2+^ for ligand recognition. CLRs can mediate TACA recognition, internalisation, antigen processing and presentation [115]. Furthermore, some of the CLRs can also induce specific signalling, which modulates myeloid cell activation and effector functions, depending on the genetic programmes that they trigger. Indeed, many CLRs contain the signalling motifs immunoreceptor tyrosine-based activation motif (ITAM) or immunoreceptor tyrosine-based inhibitory motif (ITIM) in their cytoplasmic portion, or recruit adaptor proteins that contain ITAM [118]. Other CLRs contain an hemi-ITAM motif composed of a single tyrosine within an YXXL motif [119,120]. However, some CLRs do not trigger cell signalling, although they can mediate antigen internalisation and favour antigen processing and presentation [118]. CLRs can also recognise molecules released by apoptotic or dead cells, also known as damage-associated molecular patterns (DAMPs) and might determine an immunogenic or tolerogenic immune response according to the DAMP recognition-induced signalling pathways [118,121]. Thus, myeloid CLRs are key players in maintaining immune homeostasis or activation and can modulate tumour development, growth or metastasis, as well as pathogen-specific immunity [122].

Some examples of CLRs that recognize TACAs that will be discussed in this review are: Macrophage Mannose Receptor (MMR), Macrophage Galactose/GalNAc lectin (MGL), DC-specific intercellular adhesion molecule-3–grabbing non integrin (DC-SIGN), Macrophage inducible C-type lectin (Mincle), Dendritic cell associated C-type lectin (Dectins) and DC immunoreceptor (DCIR). For instance, CLRs can contain ITAM or ITIM domains and reprogramme cell functions by inducing cytokine production and secretion, oxygen species (ROS) production, favouring antigen processing and presentation, or inhibiting Toll Like Receptor (TLR)-signalling [123]. However, certain TACAs that bind CLRs can only mediate immunoregulatory mechanisms that help tumours to evade the immune response [124]. Indeed, the incubation of immobilised galactose and mannose-based monosaccharides induces a regulatory phenotype in lipopolysaccharide (LPS)-stimulated DCs, shown by the expression of reduced levels of CD40 and increased PD-L1 and the differentiation of Tregs [125].

Last, sialic acid-binding immunoglobulin-like lectins (Siglecs) are a family of receptors that recognize sialoglycans, including those expressed by tumour cells, such as sTF, sTn and sLe^x^ antigens, among others. Siglecs mediate immunoregulatory signals both on myeloid and lymphoid immune cells [126], suppressing T cell responses [127] and NK cytotoxicity [128], inducing the expression of TGFβ by macrophages [129] and suppressing T cell responses [127]. Of note, a family of soluble lectins known as Galectins plays an important role in the modulation of the antitumor immune response in the tumour microenvironment [130]. Galectins bind to ꞵgalactosides and comprise a family with several members that can induce T cell death [131], the secretion of anti-inflammatory factors [132], immunosuppression through PD-1 [133], and angiogenesis [134]. Indeed, T-cell immunoglobulin and mucin domain-containing 3 (TIM-3) binding to its ligand, galectin-9, results in Th1 cell death [135,136]. However, we will not discuss their functions in this work, since this information can be found in recent reviews [137,138,139,140,141].

In addition to these properties, some CLRs can collaborate or inhibit other PRR-induced signalling, such as Toll-like receptors (TLRs) [142] or other CLRs [143], in a process that is commonly known as crosstalk [144,145]. For example, simultaneous triggering of DC-SIGN with TLR4 strengthens and prolongs TLR-signalling to enhance proinflammatory cytokine production in DCs [146,147]. However, Tn-glycopeptides that bind MGL modulate TLR2 triggering, favouring the production of IL-10 and TNF⍺ by DCs [142]. Furthermore, recent reports have demonstrated that expression of innate receptors can be induced by inflammatory signals, as well as by microRNAs (miRNAs) [148,149], such as DC-SIGN, MGL1 and MMR [150]. Interestingly, both DC-SIGN and MMR-induced by miR-511-3p contribute to the anti-inflammatory DC phenotype defined by increased IL-10 and PD-L1 expression that favours Th2 polarisation via upregulating IL-10 and IL-4 while inhibiting IL-17 [150,151]. Therefore, CLRs possess different immunomodulatory properties that regulate immune effector cell functions when expressed on monocytes, macrophages or DCs, among other cells. We describe below the specific mechanisms underlying TACA and membranary CLR interaction that result in the immunomodulation of the antitumor immunity (illustrated in Figure 2).

#### 4.1.1. Dectins

Dectin receptors are a group of CLRs that interact with pathogens or cancer cells [152,153]. Several independent works have demonstrated that in cancer, Dectin-1, which binds ꞵglucans in *N*-glycans, can have opposite effects in the induction of antitumor immunity. For instance, it prevents severe metastasis of melanoma or Lewis lung cancer cells by promoting antitumor effector mechanisms mediated by NK cells, DCs or macrophages in a process that depends on Interferon Regulatory Factor (IRF) 5 activation [122], favouring NK cell dependent killing of tumour cells and the recruitment of inflammatory M1 macrophages [152]. It also signals on DCs through the serine/threonine-protein kinase Raf-1 inducing the activation of nuclear factor kappa-light-chain-enhancer of activated B cells (NF-κB) and promoting the differentiation of antitumorigenic Th9 cells [154] and CTLs [155]. In addition, Dectin-1 triggering by β-glucan particles [156] or laminarin [157] suppresses the growth of subcutaneously inoculated Lewis lung cancer and gastric dysplasia, respectively, by inducing MDSC apoptosis and MHC II expression on antigen presenting cells and inhibiting angiogenesis. However, opposite results have demonstrated that Dectin-1 can also promote cancer progression. In fact, it can mediate tolerogenic macrophage programming in collaboration with galectin-9 in pancreatic cancer. Moreover, high expression of Dectin-1 has been associated with cancer and low survival rate [158]. Thus, Dectin-1 is a clear example of a CLR that plays opposing roles in tumour development depending on the microenvironment and aggressiveness of tumours (Figure 2).

On the other hand, Dectin -2, which recognizes mainly high mannosylated [159] antigens and is expressed by Kupffer cells, impedes the development of metastasis in the liver induced by colon or melanoma cells by mechanisms where Kupffer cells are able to mediate the uptake and clearance of cancer cells [160]. Therefore, as Dectin-1, Dectin-2 can induce effective antitumor immune responses. Finally, Dectin-2 can form heterodimers with the related CLR Dectin-3 (also known as Macrophage C-type lectin, MCL) [161] and contributes to the suppression of liver metastasis by enhancing the phagocytic activity of Kupffer cells [160] (Figure 2).

#### 4.1.2. MMR

MMR (CD206) is another type of CLR that is mainly expressed on DCs, macrophages and endothelial cells [162], and recognises multivalent mannosylated glycoconjugates both from pathogens and tumour cells [52]. MMR can mediate phagocytosis or endocytosis of its ligands, promoting antigen uptake and presentation to T cells [162]. In addition, MMR expressed on endothelial interacts with lymphocytes and also cancer cells promoting their trafficking [163]. Interestingly, tumours express MMR^+^ TAMs [164], suggesting a role of this molecule in tumour growth. Indeed, MMR is expressed on M2-type macrophages [164], and its expression was associated with a downregulation of IFNγ [164,165] and upregulation of IL-4 [164]. Last, MMR has also been linked to homeostasis through the clearance of hormones and self-proteins [162,163] (Figure 2).

#### 4.1.3. DC-SIGN

DC-SIGN plays an important role in homeostatic control [166], pathogen and tumour derived molecule recognition. DC-SIGN specifically binds antigens decorated with high-mannose or fucosylated Lewis-type structures [167]. As other CLRs, signalling crosstalk with TLRs has been demonstrated for DC-SIGN on DCs [168]. Interestingly, pathogen or tumour recognition by DC-SIGN on DCs often results in immune escape, transforming proinflammatory into tolerogenic signals and keeping them in an immature or tolerogenic state [169,170]. 

Interestingly, it was recently found that fucosylation of-glycans on pancreatic ductal carcinomas is associated with EMT, being high in epithelial phenotype, while absent in mesenchymal-like cells [171]. It was also proposed that the fucosylated antigens constitute potential ligands for DC-SIGN considering the fact that DC-SIGN^+^ TAMs were recruited to the tumour microenvironment [171], a process that could induce EMT (Figure 2).

#### 4.1.4. Mincle

Mincle (CLEC4E) can either associate with the FcRγ chain or with MCL (Dectin-3), after which Mincle is endocytosed promoting phagocytosis of ligands. Unlike other CLRs, Mincle is expressed on B cells apart from innate cells (macrophages, neutrophils and DCs) [172]. MINCLE recognises glycolipids from pathogens [173] and DAMPs, such as βglucosylceramide and spliceosome-associated protein 130 (SAP130) [174] as well as glycosylated and mannosylated lipids [175]. The recognition of its ligands induces the expression of several cytokines and chemokines such as TNF⍺, IL-6, MIP-2 and CXCL1 [176,177,178]. Mincle is expressed on myeloid cells from pancreatic and urothelial cancers [179,180] where it mediates immunosuppression by inducing M2 TAMs and low T cell activation [179,181]. In pancreatic cancer, Mincle triggering by SAP130 induces the recruitment of TAM and promotes tumour growth via necrosome induced-immune suppression [179]. Indeed, silencing of Mincle in tumours effectively blocked the progression of cancer [181] (Figure 2).

#### 4.1.5. DCIR

DCIR (CLEC4a) specifically recognises a broader spectrum of glycan ligands containing fucose and mannose [182,183]. It contains an ITIM, and since its stimulation inhibits the production of inflammatory cytokines, it is thought to participate in the homeostasis and control of inflammation [184,185,186]. DCIR is differentially expressed on DC depending on their origin and stage of maturation or activation state, being associated with regulatory signals and immature DCs [182]. In addition, DCIR triggering inhibits TLR9-induced IFN⍺ production by plasmacytoid DCs, although it can mediate antigen presentation by endocytosis [186]. DCIR is capable of recognizing tumour cells through sulfo-Le^a^, Le^b^ and Le^a^ antigens [183], mainly present in *N*-glycans [187] (Figure 2).

#### 4.1.6. MGL

MGL binds terminal GalNAc residues in a calcium-dependent manner and mediates various immune and homoeostatic functions [188]. Two very similar MGL isoforms have been identified in mice, mMGL1 and mMGL2, the latter being the one that presents similar carbohydrate specificity to human MGL [187,189,190]. Both expression and function of MGL have been associated with impaired or tolerogenic immune responses. Not only is MGL expressed in immature or tolerogenic DCs [191], corticosteroid-cultured macrophages [192], alternative activated macrophages or on type 2 DCs [193,194], but its triggering also dampens the immune response, by inducing the production of the anti-inflammatory cytokine IL-10 by DCs [142] and promoting the differentiation of Tregs [142]. In addition, engagement of MGL through effector T cells results in reduced proliferation and induction of T cell apoptosis, suppressing T cell activation [195].

The immunomodulatory role of MGL in cancer can be mediated by recognition of terminal GalNAc residues or the Tn antigen, which are expressed in a variety of tumour-associated proteins such as MUC1, carcinoembryonic antigen (CEA), cancer antigen 15-3 (CA15-3) and receptor tyrosine kinase c-Met, a protein that is involved in colorectal carcinogenesis, among others [196,197,198]. However, MGL ligands in ovarian tumour cells were not only restricted to mucins attached to the plasmatic membrane but also to intracellular molecules and matrix-associated components [199]. Interestingly, the presence of MGL ligands in colorectal cancer is associated with BRAF mutations and might constitute an independent prognostic marker for advanced colorectal [200,201,202] and breast cancer [84]. Finally, MGL ligand detection in tumour tissues has been correlated with cell malignancy [201,203,204,205] and has been proposed for in vivo diagnosis of tumours [206].

MGL seems to play a central role in tumour immunomodulation. Indeed Tn^+^ colorectal gliomas infiltrated with MGL^+^ TAMs are characterised by an enhanced accumulation of MDSCs and reduced levels of cytotoxic CD8^+^ T cells [207]. In addition, our group has recently demonstrated that both lung and breast tumours expressing the Tn antigen contribute to immune evasion by inducing the recruitment of MGL^+^ cells that produce IL-10 and favour the differentiation of tumour-infiltrating Tregs [124,208]. Interestingly, Tn^+^ lung tumour cells also contribute to tumour angiogenesis. Both immune regulation and angiogenesis induced by Tn^+^ tumours is dependent on MGL-expressing myeloid cells [124] (Figure 2).

Interestingly, recent data describing new MGL functions demonstrate that the tolerogenic phenotype in MGL^+^ DCs induced by the interaction between this CLR and its tumour-associated glycan ligands is associated with a metabolic quiescent phenotype in DCs that is characterised by a diminished capacity to use the glycolysis pathway for ATP generation [113].

## 5. TACAs Relevance in Tumour-Secreted eXtracellular Vesicles

As mediators of intercellular communication, extracellular vesicles (EVs), including microvesicles and exosomes, play vital roles in many aspects of cellular homeostasis, physiology, and pathobiology [209]. In the tumour microenvironment, EVs may derive from cancer cells, immune cells, and also from other non-immune host cells [209], and are key players in the crosstalk between malignant cells and the immune system [210]. Among EVs, we can distinguish microvesicles and exosomes. The main difference among microvesicles (MVs) and exosomes is that MVs are generated by the outward budding and fission of the plasma membrane, while exosomes are secreted when multivesicular bodies fuse with the plasma membrane and release their contents [211,212]. EVs are a main focus of attention in cancer research currently since they can be detected in extracellular fluids such as blood, urine and saliva, and offer a very promising approach for cancer diagnosis.

Some of the tumour-associated glycan alterations have been reported as enriched in cancer EVs, and therefore may constitute important biomarkers with the potential to be used in the clinic [213]. A comprehensive analysis of EV glycosylation compared to cell membrane glycans in human glioma cell lines showed that EVs are enriched in complex *N*-glycans, containing LacdiNAc structures with di-, tri-, and tetra antennary proximally fucosylated glycans and the presence of peripheral Galα3Gal structure, while cellular membranes were highly enriched in high mannose glycans [214]. These results outline the fact that the glycans found in EVs are not necessarily representative to those present in the glycoconjugates from the cellular membrane.

Another study that characterised the glycan profile of EVs from ovarian cancer cell lines using lectins showed the presence of glycoproteins bearing complex *N*-glycans with 2,3-linked sialic acid, fucose, bisecting-GlcNAc and LacdiNAc structures, and *O*-glycans with the TF antigen [215]. In the same line, EVs from genetically engineered gastric cancer cell lines with truncated *O*-glycosylation contained sTn [216]. Furthermore, EVs from melanoma cells and colon cancer cells express complex *N*-linked glycans, high mannose, polylactosamine, and a reduction in Blood Group A/B antigens [217], as well as the Tn antigen, suggesting that tumour-derived EVs carry TACAs similar to the parental cancer cells that produce them [217] and that they might have a role in tumour progression.

### Glycosylation Role in EV Functions

Glycosylation in EVs has a fundamental role in their uptake and signalling by immune cells [218]. Indeed, the release of N-glycans in EVs increases their uptake by different types of cells [219]. Furthermore, *N*-linked glycans can determine the EV cargo recruitment [220] by directing protein sorting to membrane microdomains within cells [221].

As far as proteoglycans are concerned, treating human cells with heparanase, an enzyme that trims long HS chains and facilitates clustering of Syndecan, has consequences on protein sorting into EVs [222]. In addition, higher levels of vascular endothelial growth factor (VEGF) and hepatocyte growth factor (HGF) in exosomes derived from heparanase high expressing cells are in comparison with heparanase low expressing cells, and better mediate spreading of tumour cells and invasion of endothelial cells [222]. Thus, proteoglycans in EVs can play an important role in tumour metastasis and angiogenesis.

Aberrant glycosylation in EVs derived from tumour cells may participate in immune modulation and antitumor immunity by interacting with CLRs, promoting metastases [210]. Indeed, breast cancer-derived EVs with bisecting GlcNAc modifications in the vesicular integrin β1 showed a diminished pro-metastatic function in recipient cells [223], and *N*-glycosylation in the I-like domain of this protein was found to be essential for EV-mediated metastasis but not proliferation of recipient cells [224]. In prostate cancer, increased cellular expression of FUT8 reduces the secretion of EVs but increases abundance of proteins associated with metastasis in these vesicles and alters the glycans expressed on the EV-derived glycoproteins [225]. Finally, in colon cancer cells with abnormal *O*-glycosylation, the percentage of CD44 in exosomes is significantly higher than in those derived from normal cells [226]. CD44 is related to many pathophysiological features of carcinomas due to its regulation in signalling pathways, and in this study it is shown that aberrant *O*-glycosylation can regulate expression or delivery of this protein, as *O*-glycan truncated CD44 was mainly released via exosomes instead of participating in cellular biological activities [226].

Finally, there is promising evidence that exploring aberrant glycosylation in tumour-derived EVs may lead to the identification of novel biomarkers for cancer diagnosis. For instance, CD63, an EV marker overexpressed in breast cancer, may be detected in serum through fucose binding lectins [227].

## 6. TACA and CLR Targeting for Cancer Immunotherapy

Considering the role of TACA in cancer progression, together with the expression and function of CLRs, targeting antigen presenting cells (such as DCs and macrophages) using CLR ligands or specific antibodies is an attractive strategy to recruit and activate these cells. Indeed, specific glycan ligands can activate or suppress downstream signalling of a certain CLR that might regulate DC maturation, migration, antigen uptake, processing, and presentation as well as cytokine production influencing adaptive immune responses [228]. However, the immunomodulatory properties of the targeted CLR must be considered to secure the induction of an inflammatory antitumor immune response. Sometimes, the immune regulatory properties of CLRs can be overridden when they are targeted in combination with adjuvants such as TLR ligands [229,230] or TACA-specific antibodies [231,232]. In some cases, the induction of tolerance can be reverted with coinjection of anti-CD40 antibodies to activate DCs and to inhibit tumour growth [233,234].

When targeting CLR with glycan ligands, several properties of the ligand have to be taken into consideration, such as the nature and multivalency of the antigen to promote crosslinking. The use of artificial against natural glycan ligands may increase its immunogenicity. In addition, both the spatial orientation of the ligand and the multivalency can enhance affinity of the glycan to CLRs. Specific glycans can be conjugated to tumour antigen peptides, proteins or DNA, incorporated in nanoparticles or liposomes, or even into adenoviruses and lentiviral vectors. On the other hand, antibodies can be conjugated to a variety of antigen formulations. Multiple CLRs show potential in antitumor DC-targeted strategies. Here, we will focus on CLRs that are expressed in the tumour microenvironment and recognize TACAs. We will not, however, discuss the efficacy of TACAs as anti-tumor vaccines [17] but rather as tools to target CLRs. Passive immunotherapy with TACA-specific antibodies or strategies that inhibit Siglecs are reviewed elsewhere [26,235].

MMR in vivo targeting has been extensively studied. Mannose-rich glycan structures conjugated to liposomes [236,237], polyamidoamine dendrimers [238] or polylactide-co-glycolide (PLGA) based nanoparticles [239], results in increased IL-12 with IgG antibodies, enhanced CD4^+^ T and CD8^+^ T cells and protection against tumour upon immunisation [240]. Initially, mannosylated conjugated MUC1 generated both antitumor B and T cell responses in preclinical and clinical studies [241,242]. More recently, MMR-targeting was proposed for TAM modulation with siRNA [243] or DNA-based vaccines [244] to favour photothermal tumour therapy [245], or as carriers of chemotherapeutic drugs [246]. In particular, a recent study showed that mannose-functionalized nanoscaffolds as carriers of chemotherapeutic agents [247] efficiently killed tumours, although more in vivo studies are needed to confirm their efficiency. However, the expression of MMR in other cell types other than macrophages and immature DCs, such as endothelial cells, retinal pigment epithelium, kidney mesangial cells, and tracheal smooth muscle cells [164], makes the in vivo MMR-targeting a challenge.

On the other hand, DC-SIGN targeting using glycoconjugated molecules has also been used to deliver cancer antigens to DCs in order to enhance uptake and antigen presentation for the development of more effective cancer immunotherapies [248,249,250]. Indeed mannosyl-based dendrimers, liposomes or nanoparticles [229,251,252,253,254] or even tumour-specific highly mannosylated peptides conjugated with a TLR7 ligand [229], successfully achieved multivaling presentation and enhanced binding to DC-SIGN. However, it should also be considered that binding affinity to DC-SIGN does not always enhance antigen presentation [229]. Interestingly, tumour-derived EVs can target DC-SIGN expressed on DCs. Indeed, upregulation of high mannose glycans in the tumour cell surface enhanced the EV uptake by DCs through DC-SIGN, facilitating the priming of tumour-specific CD8^+^ cells [255].

DC-targeting through CLR can also enhance immune responses through B cell differentiation allowing the secretion of high levels of antitumor antibodies. For instance, since Dectin-1 is mainly expressed by mouse conventional type 2 DCs that favour CD4^+^ Th differentiation, its targeting with a specific antibody induces strong CD4 T cell differentiation and B cell responses [256]. Furthermore, Dectin-1 targeting can also induce activation of human DCs and antigen-specific CD8^+^ T cells [257]. In the context of cancer, Dectin-1 targeting using free ꞵglucans [258,259,260] and ꞵglucan-coated gold nanoparticles carrying a MUC4-TF peptide [261] induces strong humoral responses, demonstrated by the presence of high titres of specific antibodies and antigen-recognizing T cells [261]. However, the ability of these antibodies to inhibit tumour growth was not explored, indicating the need to prove their antitumor efficacy.

In the same line, the use of glycosylated dendrimeric molecules carrying the Tn antigen has been successfully used to target dermal DCs through MGL2, favouring capture and processing of glycosylated Tn antigen in vivo and inducing a MHC II-restricted T cell response and expansion of the germinal centre B-cells, therefore generating a strong antitumor immune response [194]. In addition, MGL1 in vivo targeting has been successfully achieved using galactose conjugated PLGA-polymers in an experimental model of colitis [262]. Recently, a multivalent peptide mimetic of GalNAc was reported to inhibit ovarian cancer through MGL2-targeting [263].

As with Dectin-1 and MGL2, DCIR2-targeting with a specific antibody revealed that this receptor induces preferentially CD4^+^ Th2 cell differentiation [264] and robust B cell responses [265]. However, no targeting to this CLR has been performed with glycan ligands, probably due to its broader specificity of fucosylated and mannosylated glycans than other CLRs.

Last, only one report of in vivo targeting to Mincle is available, where the immunomodulatory properties of mannosylated and glucosylated lipid antigens were evaluated [175]. Interestingly, glucosylated ligands resulted in significantly increased levels of IL-2, IFNγ, and IL-17 by splenocytes, while no differences with the control were found when mannosylated ligands were injected. Importantly, the effect of glucosylated ligands was abrogated in Mincle-knockout mice [175], although it is unknown which cells are playing a role in the induced immune response.

## 7. Novel Immunotherapeutic Strategies Based on TACAs

The identification of the functions that mediate TACAs on cancer cells has led to the development of clinical approaches to treat cancer. A great number of strategies have been developed over the three past decades, though, the majority did not reach FDA approval. However, specific monoclonal antibodies against Tn or sTn-MUC1 and MUC16 glycopeptides and GD2 (dinutuximab) have been approved by the Food and Drug Administration (FDA) for their use in cancer treatment [266,267,268,269].

New TACA-based approaches for antitumor immunotherapy are being studied at the moment. These include the engineering of chimeric antigen receptor (CAR) T cells and the production of bispecific antibodies [26,270]. Adoptive transfer of CAR T cells is a promising immunotherapy strategy to treat cancer in an MHC-independent manner that grants activation, proliferation and survival of specific adoptively transferred T cells [271]. GD2-specific CAR T cells have been demonstrated to reach solid tumour sites [272] and to produce IFNγ that reprogrammed the tumour microenvironment [273]. However, intratumoral CAR T cells expressed PD-1, which can limit antitumor efficacy [273]. In fact, the combination of CAR T based immunotherapy with anti-PD1 blockade favours CAR T cell survival [274]. On the other hand, anti-sTF-MUC1 peptide CAR T cells were also developed [275] and reported to induce apoptosis and necrosis of tumour cells [276] associated with the production of proinflammatory cytokines and delaying tumour growth [275].

Bispecific antibodies are engineered antibodies that can simultaneously recognize two different epitopes [277]. However, limited studies provide evidence of the potential of TACA-based bispecific antibodies in cancer. Anti-GD2 and CD3 bispecific antibodies have been successful in tumour cell killing [278,279] and reduced tumour growth by increasing T cell activation and proliferation mediated by monocytes [279]. Similarly, anti-MUC1 and CD3 bispecific antibodies were also capable of inhibiting tumour growth [280], although they have been significantly less studied than anti-GD2 bispecific antibodies [275].

## 8. Conclusions and Perspectives

TACAs expressed by tumour cells or their derived-EVs, together with CLRs present in innate leukocyte, possess immunomodulatory properties that can favour tumour growth by different molecular and cellular mechanisms. Furthermore, CLRs can be targeted to induce effective antitumor immunity, either using specific antibodies or TACAs. However, the latter remains a challenge due to the broad ligand specificity of some CLRs. In this sense, the elucidation of their role in tumour-induced immune evasion is essential to develop antitumor immune therapeutic strategies. Furthermore, the dual targeting of CLRs combining two different TACA-based molecules could improve the desired antitumor immune response. In some cases, the targeting of CLR together with a proinflammatory stimulus, such as TLR ligands, is essential to ensure an antitumor response, since many CLRs mediate tolerogenic programmes on antigen presenting cells. Indeed, CLR-targeting to induce antigen-specific tolerance has been recently proposed as a possible strategy to treat inflammatory or autoimmune diseases [281]. Finally, some approaches that could be exploited in the near future need to be further investigated, such as CLR targeting in combination with checkpoint blockade, through the use of anti-PD1 or anti-CTLA4 antibodies, as well as CLR targeting combined with immunogenic chemotherapy. Future research in these areas will certainly provide new insights for CLR-based cancer immunotherapy using TACAs as well as on their roles in cancer.

## Figures and Tables

**Figure 1 cancers-14-01854-f001:**
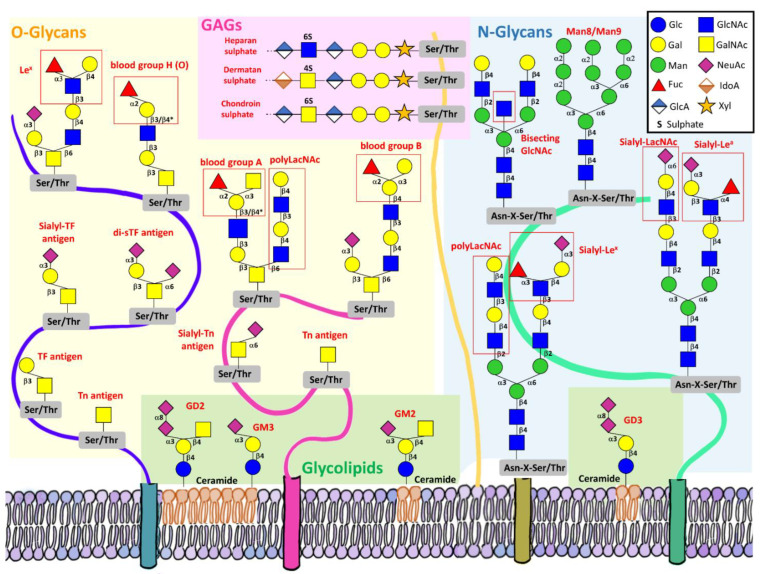
Tumour-associated glycoantigens expressed in adenocarcinoma cells. TACAs present in *O*-glycoproteins (yellow background), *N*-glycoproteins (blue background), GAGs (purple background) and glycolipids (green background) are shown. Their usual names are written in red.

**Figure 2 cancers-14-01854-f002:**
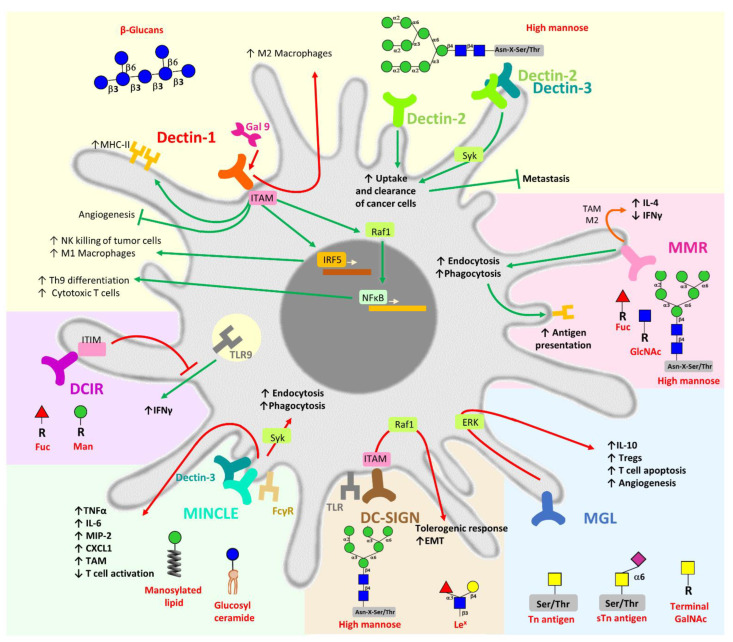
TACA recognition by CLRs in cancer. The role of CLRs in TACA recognition is shown: Dectins (yellow background), MMR (pink background), MGL (blue background), DC-SIGN (brown background), Mincle (green background) and DCIR (purple background. Effect of TACA-induced CLR signalling is shown. In red and green lines, the protumorogenic immunomodulatory processes and the antitumor effects, respectively, are represented. Carbohydrate symbol legend is shown in Figure 1.

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
