# Peer review of "Advances in the Immunomodulatory Properties of Glycoantigens in Cancer"

_cancers, 2022, doi:10.3390/cancers14081854_

Round 1

Reviewer 1 Report

In this review manuscript entitled “Advances in the immunomodulatory properties of glycoantigens in cancer”, da Costa and Freire summarized tumor-associated carbohydrate antigens (TACAs) from the aspects of their biosynthetic alterations in cancers, their immunomodulatory roles through the interactions with endogenous lectins (e.g. C-type lectin receptors (CLRs)), and tumor-secreted extracellular vesicles (EVs). The authors then described targeting TACAs and CLRs for cancer immunotherapy including novel TACA-based approaches.

If the authors were to submit again, I think they should revise the whole structure. Section 5 onwards should be summarized mainly because it is relatively true to the title of the manuscript. In Section 5 onwards, many relatively new original papers are cited compared to the previous sections. This will be useful for this reader.

Since this review manuscript is about tumor-associated glycans. I think that it is a good idea to start with an introduction to the types of glycans. However, the content of the review is just a list of what is written in glycobiology textbooks, and most of the references are to review articles. The authors should be more concise in describing the contents up to Section 4. Generally speaking, in review articles, the original works should be cited as much as possible so that the readers can easily confirm the contents. What is even worse is that there are places in the description that are considered to be erroneous or misleading. In Figure 1, the authors refer to GAGs as glycolipids. In the second paragraph of Section 1, the authors describe that the biosynthesis of O-glycans occurs in the Golgi apparatus, but some occur in the endoplasmic reticulum. In the third paragraph, the authors describe that the biosynthesis of GAGs begins in the cytoplasm, but which step of GAGs biosynthesis occurs in the cytoplasm specifically? In the seventh paragraph of Section 1, the authors describe the glycosylation of E-cadherin, but has it been proven that the function of E-cadherin is regulated by glycosylation?

Author Response

In this review manuscript entitled “Advances in the immunomodulatory properties of glycoantigens in cancer”, da Costa and Freire summarized tumor-associated carbohydrate antigens (TACAs) from the aspects of their biosynthetic alterations in cancers, their immunomodulatory roles through the interactions with endogenous lectins (e.g. C-type lectin receptors (CLRs)), and tumor-secreted extracellular vesicles (EVs). The authors then described targeting TACAs and CLRs for cancer immunotherapy including novel TACA-based approaches. If the authors were to submit again, I think they should revise the whole structure. Section 5 onwards should be summarized mainly because it is relatively true to the title of the manuscript.

Response: According to the reviewer´s comment, we assume that he asks us to resume the sections before section 5. Thus, on sections 1, 2 and 3, we have reorganised and summarised the information to focus this review on tumour associated glycans. Nevertheless, we feel some general information about glycan structure is necessary for the complete understanding of non-specialized readers on this topic, so we made some short descriptions of each type of glycan structure before describing the specific TACAs.

In Section 5 onwards, many relatively new original papers are cited compared to the previous sections. This will be useful for this reader.

Response: On sections 7 and 8 we have cited both reviews and original papers. The reviews were cited usually at the beginning of the section to give a general comment on the subject. Then, when we described particular results we cited the appropriate research or original papers. As you can see the following reference numbers (corresponding to the submitted manuscript) are original papers: 117, 119, 127, 128, 131, 132, 136, 138, 139,141-145, 147, 148, 152, 153, 156-161, 163-167, 169-175, 177-179, 181-183, 185-196, 202-205, 207-209, 212-215, 217-222, 224-230, 232-248, 256-258, 262-263. When the finding is relatively old, we prefer to cite the original paper instead of citing a recent review. As the reviewer will see, both sections contain these types of references. However, we have considered the reviewer's comment and included some other original papers at the beginning of section 4.1 such as 120 and 121 (reference numbers from the revised manuscript). We hope that the reviewer finds our response suitable, and apologize in advance in case we have misunderstood the reviewer.

Since this review manuscript is about tumor-associated glycans. I think that it is a good idea to start with an introduction to the types of glycans. However, the content of the review is just a list of what is written in glycobiology textbooks, and most of the references are to review articles. The authors should be more concise in describing the contents up to Section 4. Generally speaking, in review articles, the original works should be cited as much as possible so that the readers can easily confirm the contents.

Response: Response 1 and rearrangement of some sections takes this comment into account.

What is even worse is that there are places in the description that are considered to be erroneous or misleading. In Figure 1, the authors refer to GAGs as glycolipids.  In the second paragraph of Section 1, the authors describe that the biosynthesis of O-glycans occurs in the Golgi apparatus, but some occur in the endoplasmic reticulum. In the third paragraph, the authors describe that the biosynthesis of GAGs begins in the cytoplasm, but which step of GAGs biosynthesis occurs in the cytoplasm specifically? In the seventh paragraph of Section 1, the authors describe the glycosylation of E-cadherin, but has it been proven that the function of E-cadherin is regulated by glycosylation?

Response: We would like to thank the reviewer for the observations and apologise for the mistakes. We have corrected figure 1 and gags/glycolipids descriptions (please refer to section 1.4 and 1.5). Furthermore, we clarified and added information about the synthesis of GAGs (section 1.5) (pages 4 and 5). Although, to our understanding, there is no report associating glycosylation to E-cadherin function, it is documented that glycosylation is fundamental for E-cadherin localization in the cell surface, and therefore to cellular adhesion. This was clarified in section 2.2. Modified text is shown in red on page 6.

Reviewer 2 Report

The review titled “Advances in the immunomodulatory properties of glycoantigens in cancer” by Costa and Freire is a well written review which summarises how aberrant glycans influence tumour growth and immunity, their application in cancer treatment and spotlights. It is a well drafted overview on the topic and can be accepted after some minor corrections.

Check for spelling errors as in the legend of the Fig 1. on page 3, “… writtehn in red”

Where is section 7?

Although the section 8 deals with the novel immunotherapeutic strategies based on TACAs, the review does not discuss the status quo of the glycans or glycoantigens in the clinical use. Is there any FDA approved therapy? How about clinical trials are there any?

Author Response

The review titled “Advances in the immunomodulatory properties of glycoantigens in cancer” by Costa and Freire is a well written review which summarises how aberrant glycans influence tumour growth and immunity, their application in cancer treatment and spotlights. It is a well drafted overview on the topic and can be accepted after some minor corrections.

Check for spelling errors as in the legend of the Fig 1. on page 3, “… writtehn in red”

Response: The spelling error has been corrected.

Where is section 7?

Response: Section 7 was placed between sections 6 and 8 of the submitted version of the review. We invite the reviewer to comment on this section.

Although the section 8 deals with the novel immunotherapeutic strategies based on TACAs, the review does not discuss the status quo of the glycans or glycoantigens in the clinical use. Is there any FDA approved therapy? How about clinical trials are there any?

Response: According to the reviewer´s suggestion, we have a paragraph about passive immunotherapy strategies based on TACA recognition that have been approved by FDA at the beginning of section 7. Please, see new text in red on page 16.

Reviewer 3 Report

The authors provide an overview of the rapidly evolving area of glycosylation and its role in tumor immunity and emerging immunotherapies. Figure 2 and the sections describing the C-type lectin receptors are well done and are the main core of the review. There are some omissions and corrections that need to be addressed, as follows:

  • Figure 1 does not describe GAGs correctly, nor does the text in section 2.4. What is shown are glycosphingolipid ganglioside species.  The description and depiction of glycosaminoglycan polymers needs improving, with emphasis on highlighting the major types of GAGs associated with cancers (heparan sulfate and chondroitin sulfate) and their protein carriers, proteoglycans. These just need to be overall summarized better in the text.  Attention to glycosphingolipid descriptions is also warranted, as there is no section like 2.4. for example.
  • There is an omission of the description of the soluble immune lectins (galectins, siglecs) and selectins associated with many tumor types. These should at least get a summary paragraph
  • Minor edit suggestion. There are multiple papers from Brian Haab's laboratory in the past four years describing a new sialic acid tumor antigen in pancreatic cancer. The antigen is similar to sLeA and sLeX except there are no fucose. It is one of the few new carbohydrate antigens discovered in the genomic era. This could be mentioned in section 2.3.

Author Response

The authors provide an overview of the rapidly evolving area of glycosylation and its role in tumor immunity and emerging immunotherapies. Figure 2 and the sections describing the C-type lectin receptors are well done and are the main core of the review. There are some omissions and corrections that need to be addressed, as follows:

Figure 1 does not describe GAGs correctly, nor does the text in section 2.4. What is shown are glycosphingolipid ganglioside species.  The description and depiction of glycosaminoglycan polymers needs improving, with emphasis on highlighting the major types of GAGs associated with cancers (heparan sulfate and chondroitin sulfate) and their protein carriers, proteoglycans. These just need to be overall summarized better in the text.  Attention to glycosphingolipid descriptions is also warranted, as there is no section like 2.4. for example.

Response: We apologise for the mistake. As suggested by the reviewer, be have corrected figure 1 and the text regarding GAGs. We also added a section to describe glicosphingolipids (please see section 1.4) and expanded the description of GAGs associated with cancer (please refer to section 1.5).

There is an omission of the description of the soluble immune lectins (galectins, siglecs) and selectins associated with many tumor types. These should at least get a summary paragraph.

Response: We would like to acknowledge the reviewer for this comment and point out that there was a paragraph on page 6 briefly describing Siglecs. As suggested by the reviewer, information on galectins have been added on pages 9 and 10. Selectins are defined in a small paragraph added on page 6 in the subsection 2.2.

Minor edit suggestion. There are multiple papers from Brian Haab's laboratory in the past four years describing a new sialic acid tumor antigen in pancreatic cancer. The antigen is similar to sLeA and sLeX except there are no fucose. It is one of the few new carbohydrate antigens discovered in the genomic era. This could be mentioned in section 2.3.

Response: We would like to acknowledge the reviewer for this comment and as suggested, it was added in section 2.3 (now 1.3 after some restructuring). Please see new text in red on page 3.

Round 2

Reviewer 1 Report

The authors have addressed and revised almost all of the concerns I have presented.

However, I would like to point out three points in the newly added picture of GAGs in Figure 1.

The first is that if the authors show the core linkage-region tetrasaccharides of GAGs chains, the fourth sugar in the same region of DS should be glucuronic acid since they are identical.

Second, it would be nice if sulfation, which is mentioned in the text in the new section 1.5, could also be represented in the diagram.

Third, the widely recommended symbol for iduronic acid is to paint the bottom half brown, not the top half. The authors may want to check out https://www.ncbi.nlm.nih.gov/pmc/articles/PMC7335484/.

Once these points are improved, I would recommend that the paper be accepted.

Author Response

We thank the reviewer for his/her helpfull suggestions. Figure 1 has been corrected accordingly.